# Detection of Mast Cells in Human Superficial Fascia

**DOI:** 10.3390/ijms241411599

**Published:** 2023-07-18

**Authors:** Caterina Fede, Lucia Petrelli, Carmelo Pirri, Cesare Tiengo, Raffaele De Caro, Carla Stecco

**Affiliations:** 1Department of Neurosciences, Institute of Human Anatomy, University of Padova, Via Gabelli 65, 35121 Padova, Italy; caterina.fede@unipd.it (C.F.); lucia.petrelli@unipd.it (L.P.); carmelo.pirri@unipd.it (C.P.); rdecaro@unipd.it (R.D.C.); 2Plastic and Reconstructive Surgery Unit, University of Padova, Via Giustiniani, 2, 35128 Padova, Italy; cesare.tiengo@unipd.it

**Keywords:** superficial fascia, mast cell, inflammation, tissue healing

## Abstract

The recent findings showed that the superficial fascia is a fibrous layer in the middle of hypodermis, richly innervated and vascularized, and more complex than so far demonstrated. This study showed the presence of mast cells in the superficial fascia of the human abdomen wall of three adult volunteer patients (mean age 42 ± 4 years; 2 females, 1 male), by Toluidine Blue and Safranin-O stains and Transmission Electron Microscopy. Mast cells are distributed among the collagen bundles and the elastic fibers, near the vessels and close to the nerves supplying the tissue, with an average density of 20.4 ± 9.4/mm^2^. The demonstration of the presence of mast cells in the human superficial fascia highlights the possible involvement of the tissue in the inflammatory process, and in tissue healing and regeneration processes. A clear knowledge of this anatomical structure of the hypodermis is fundamental for a good comprehension of some fascial dysfunctions and for a better-targeted clinical practice.

## 1. Introduction

The superficial fascia is a dense collagenous connective tissue layer in the middle of hypodermis, dividing the superficial adipose tissue (SAT) from the deep adipose tissue (DAT). It was described for the first time in the anterior abdominal wall by Antonio Scarpa [1,2] and recently visualized in many other parts of the body [3,4,5,6]. In recent years, the involvement of the superficial fascia in numerous regulating and modulating functions has been demonstrated, thanks to the presence of a rich vascular pattern and a huge network of nervous structures that supply this fibrous layer. The vascular network of the Scarpa’s fascia is well-branched, with a homogeneous spatial distribution, and it is composed of arteries, veins, capillaries and lymphatic vessels [7], with a percentage of immunoreactive lymphatic structures (IR%) equal to 31.2 ± 14.1% [8]. Moreover, the tissue is pervaded by nerve fiber bundles around blood vessels and adipocytes, but also penetrating the connective tissue itself, with a percentage of autonomic innervation close to 34%, highlighting the role of the superficial fascia in thermoregulation, exteroception and pain modulation [9].

Furthermore, among the different functions attributed to the subcutaneous tissue, its involvement, together with inflammatory transient cells, in the early inflammatory stages, in tissue healing and regeneration process has to be considered [10,11]. Tissue repair and renewal is a complex biological reaction involving immune cells, which, together with connective tissue cells and several humoral factors, accomplish sequential phases of hemostasis, inflammation and proliferation, to restore damaged tissue and permit the tissue healing. In these processes, mast cells play an important role as mediators of inflammation and immunoregulation. They are functionally and phenotypically heterogeneous cells, depending on the microenvironment in which they mature; their activation can induce the release of pre-formed mediators from their granules, and the synthesis of new mediators, cytokines and chemokines [12].

Therefore, the aim of this study was to analyze the presence of mast cells in the superficial fascia of the abdomen wall; the early evidence described only fibroblasts as the cell population of the superficial fascia, but all the recent findings lead us to believe that this fibrous layer is more complex than previously imagined. So far, no clear evidence shows in this structure the presence of mast cells; this study aims, for the first time, to show their presence, localization and density in the layers of human superficial fascia.

## 2. Results

The hematoxylin and eosin stain of full thickness specimens (from skin to SF, Figure 1A) showed how superficial fascia (SF) divides the superficial adipose tissue (SAT), organized in fat lobules with evident fibrous septa (retinacula cutis superficialis, RC), from the deep adipose tissue (DAT).

The Safranin-O stain demonstrated the presence of various mast cells (indicated by the arrows, Figure 1B) in the superficial fascia layer. They are uniformly distributed in the tissue and show intense staining; at higher magnification, it is evident the presence of highly stained granules in the cytoplasm of the mast cells (Figure 1C). This shows the typical dense granular cytoplasm, with a central nucleus, often obscured by the granules.

The SF was then isolated from the SAT and the DAT (Figure 1D): the tangential sections of the flat embedded SF stained by Toluidine Blue demonstrated, as already described above, a high presence of mast cells in the superficial fascia layer (Figure 1E,F, indicated by the arrows). The higher magnification image (Figure 1F) shows both elongated and rounded mast cells between the fibroblasts and the collagen fibers (c), with blue/purple metachromatic-stained cytoplasmic granules.

Mast cells are ubiquitously distributed in the SF tissue: they are evident between the collagen fibers (Figure 2D–F), near the wall of blood vessels supplying the SF (Figure 2A–C) and close to the nerve fibers crossing the fibrous connective tissue of the SF (Figure 2G–I). Figure 2G–I show nervous structures, positive to S100 immunostaining for myelinating and non-myelinating Schwann cells (brown immunoreaction), and demonstrate the presence of mast cells positive to Toluidine Blue, close to the nerves. Panel C indicates the α-sma-positive smooth-muscles cells of the arteries supplying the SF, and Panel F shows the immunoreaction for Collagen I. In all the images are evident mast cells (indicated by the arrows).

The average density of mast cells calculated by computer-assisted image analysis with ImageJ software was equal to 20.4 ± 9.4/mm^2^ (Figure 3A–C). The manual count by optical microscopy of Toluidine Blue images confirmed that the mast cells are distributed in all the tissue (Figure 3D). Most of these cells (51 ± 9.7%) are in the connective tissue area in the midst of the collagen fibers (*p*-value < 0.001), whereas 25 ± 12.7% are close to the nervous fibers, and 24 ± 4.9% are close to the blood vessels and the adipocytes, in the loose connective tissue of SF (Figure 3D).

Figure 4 shows details of the presence of mast cells close to the blood vessels (Figure 4A,C) and between the fibroblast and the collagen fibers (Figure 4B,D), by both Safranin-O staining (Figure 4A,B) and Toluidine Blue (Figure 4C,D). Mast cells differ from the fascial fibroblasts by the shape (usually more rounded) and by the intensity of the stainings: granules are intensively stained in all the cytoplasm, in red by Safranin and in purple by Toluidine Blue, respectively. Granules also cover the nucleus area.

Semithin sections of Figure 5 offer additional details of the SF structure in which we found mast cells: around the collagen fibers (c) are present some elastic fibers (e), some adipocytes (a) and blood vessels (v) (it is visible the lumen with erythrocytes). Between all these structures, some mast cells (indicated in Figure 5 by the asterisks) can be found: Toluidine Blue stainings of the semithin sections permitted highlighting the nuclei of the mast cells and a huge amount of secretory granules in the cytoplasm (intensively blue-stained). These intracellular secretory granules are also evident with Transmission Electron Microscopy (Figure 6): they are densely packed, and they fill the cytoplasmic area of mature mast cells; most of them have large dimensions and electron-dense content. Small areas of intergranular cytoplasm are visible; they contain relatively unspecialized cytoplasm with rough endoplasmic reticulum. Figure 6C,D show two mast cells with evident surface projections, between areas of densely packed collagen fibers transversally oriented (co). Figure 6A shows one mast cell in close contact with a fascial fibroblast: these cells are near to collagen fibers (co), adipocytes (a) and one unmyelinated nerve fiber, whose axons and details are visible in panel B, at higher magnification.

## 3. Discussion

A first evidence of mast cells in human fascia was shown in the deep fascia of the lower limbs [14], as well as in human fascia lata [15], with Transmission Electron Microscopy. Furthermore, a recent work highlighted the presence of mast cells in the superficial fascia of rats, closely associated with fascial adipogenic progenitors and mature adipocytes: the numerous heparin-loaded granules released from mast cells were distributed around fascial preadipocytes, indicating that mast cells could serve as endogenous physiological factors to initiate fascial adipogenesis [16]. However, this work constitutes the first histological investigation of the presence of active mast cells in the human superficial fascia, ubiquitously distributed, especially between the collagen fibers of the tissue (~51%).

Mast cells are immune cells of the myeloid lineage, present in connective tissues throughout the body, tendentially concentrated near the blood vessels, in the glands of some tissues, and some cells are randomly dispersed in the tissue [17,18]. They present a high content of secretory cytoplasmic granules and, when appropriately activated, undergo degranulation, a process by which the granules content is rapidly released into the surroundings. The effects that mast cells have are closely associated with the biological actions of the granule compounds that they release [19]. Some of the substances contained in the granules can be histamine (with vasodilator action), heparin (with anticoagulant action), serotonin, dopamine and lysosomal enzymes, but also proteases, proteoglycans, cytokines and growth factors (TNF, VEGF, TGFβ, nerve growth factor, IL-6).

The finding of the presence of active mast cells in SF, distributed all over the tissue, but above all, in the dense connective tissue itself, with a very statistically significant difference (*p* < 0.001) with respect to both nervous fibers and vessels/adipocytes areas, can help to suppose that the SF participates in regulating and healing processes. Indeed, if the role of mast cells was first recognized only associated with allergic phenomena and pathological response to antigens IgE mediated, now it is clearly demonstrated that the mast cells play a fundamental role in tissue healing processes.

Kennelly and coauthors [20] have shown the significant role of mast cells in the early inflammatory stages of wound healing in the skin, explaining their role in influencing the skin tissue proliferation and remodeling, as well as their role as modulator of intestinal healing after surgical interventions, both in humans and in animal models. Some mast cells were found also in the cardiac epithelium, stimulating not only the synthesis and store of the adrenomedullin as a potent vasodilatory hypotensive peptide [21], but also the expression of adhesions molecules, which enhance the neutrophil transmigration in ischemic cardiac tissue, favorable for tissue regeneration [22].

Some authors demonstrated that mast cells have different roles according to the different stages of wound healing, modulating the content and secretion of the granules. After the initial tissue injury, they release proteases to break down the extracellular matrix; then, they can regulate the tissue regeneration by the secretion of prostaglandins and tryptase, and consequently, after 4–7 days, they can produce VEGF, histamine and other factors, stimulating angiogenesis and wound contraction [23]. Consequently, a surplus or a deficit of mast cells mediators can conduct to keloids and hypertrophic scars or to the delayed closure of wounds and the transition of acute to chronic inflammation [24]. However, the distribution of mast cells changes according to age (it is higher in children with respect to adults) and shows some heterogeneity, considering also the fact that mast cells are frequently sparse and can be easily missed during the manual count [25]. However, we found a density equal to 20.4 ± 9.4/mm^2^, which is in line with what has already been shown: the normal range of mast cells in adult skin (mean age 43.8 ± 16.1 years) was established for some authors below 40/mm^2^ [25]. Other works showed how the density increases in the skin of patients with early stages of scleroderma (111 +/− 28 cells/mm^2^) with respect to normal controls (50 ± 14 cells/mm^2^) [26]. It has to be considered that the differences among regions might be due to region-specific mast cell roles: some authors, for instance, showed a mast cell richness in the muscle coat in the human ileocecal region, especially in the inner circular muscle layer (density equal to 72.83/mm^2^), which might be important in regulating its motility [27]. The majority of published data support a pro-fibrotic role for mast cells in the skin and connective tissue, but recent studies have showed a regulatory role in the fibroblast activity, by secreting mediators that act in a paracrine manner or by a direct communication with fibroblasts through gap junctions [28]. This evidence can be supported by our findings of mast cells in direct contact with the fascial fibroblast, opening the possibility of the involvement of the SF in regulating and modulating functions. It was recently demonstrated the active role of the SF in wound healing and skin repair. Correa-Gallegos and coauthors [29] showed that skin scars originate from the SF: in particular, sentry fascia fibroblasts rise to the skin surface after wounding and preassemble together cells and matrix components, including blood vessels, macrophages and peripheral nerves, to form a provisional matrix, needed for the heal wounds. These new findings demonstrated the ability of the SF to mobilize its tissue assemblage [30], opening new perspectives for wound repair in diabetes and fibrous diseases. Other works already demonstrated that the SF’s mechanical properties undergo harmful modifications in diabetic patients [31,32,33]: the SF becomes thicker, stiffer and harder, with an alteration of force distribution and a consequent alteration of microcirculation, and triggers activation through the spinal cord and local reflex with the skin.

Again, the more superficial fascia is studied, the more it is discovered that it encompasses multiple functions. The first evidenced role was to guarantee autonomy between the skin and the muscle/deep fascia [3,34]. But the SF works also to create a sort of elastic containment for nerves and vessels [35]. Moreover, it shows a rich vascularization [7] and hosts a lymphatic plexus [8], with important implications for reconstructive surgery and pain management of clinical manifestations caused by altered lymphatic transport (like lymphedema) [36,37,38] or circulation problems. Lastly, the SF has a confirmed sensory role thanks to the rich thin innervation supplying the tissue, with ~34% of autonomic nervous fibers [9], which can be influenced by a stress condition or a sudden change in temperature [39]. All this evidence highlights the complex and fascinating role of the SF that should be considered in clinical practice for a good comprehension of some fascial dysfunctions and for a better-targeted clinical practice.

Surely, this work present some limitations. First, we have to consider that the volunteer patients are all obese, because they are patients that underwent abdominoplastic surgery in Padova Hospital. Further studies should add some controls with normal BMI range, to avoid alterations of the superficial fascia tissue of obese patients; however, we decided to not consider the possibility to collect samples from cadavers because mast cells can degranulate during the conservation methods. Furthermore, future studies will shed light on the content of the granules of mast cells; a further analysis will be fundamental for the classification of these mast cells into the two classical subsets, tryptase-positive and chymase-positive [40]. This characterization will permit a better comprehension of their role and function and of the activated pathways in the superficial fascial tissue. However, this work constitutes the first demonstration of the presence of mast cells in human superficial fascia, adding a new comprehension of the role of this connective tissue layer in tissue healing, inflammation and treatment of fascial dysfunctions.

## 4. Materials and Methods

### 4.1. Sample Collection

Samples of superficial fascia (SF, 500–700 µm thickness) were harvested from the subumbilical region of three adult volunteer patients (mean age 42 ± 4 years; 2 females, 1 male), who were recruited at the Plastic and Reconstructive Surgery Unit of the University of Padova Medical Center for abdominoplastic surgery. The ethical regulations regarding research on human tissues described were carefully followed, in accordance with the rules described by Macchi and coauthors [41]. According to Italian law, the parts of the body removed for therapeutic purposes during surgery that would otherwise be destined for destruction can be used for research. All subjects participating in the study received a thorough explanation of the risks and benefits of inclusion and gave their oral and written informed consent to publish the data.

Given the large anatomical area operated on for abdominoplasty, 8–10 specimens (approximately 1 × 1 cm) for each subject were randomly collected in different areas of the abdominal region. The superficial fascia layers were isolated from the surrounding adipose tissue and fixed in 10% buffered formaldehyde, pH 7.4; dehydrated in graded ethanol and in xylene; and embedded in paraffin. Then, 5 μm tangential sections were cut by microtome, dewaxed and hydrated for histological and immunohistochemical stains.

### 4.2. Toluidine Blue and Safranin-O Staining

Dewaxed sections were stained in a 0.1% Toluidine Blue (Merck, Darmstadt, Germany) solution in distillate water with 0.75% acetic acid for 5 min, or in 0.5% Safranin-O (Merck, Darmstadt, Germany) solution in acetic acid 3% for 10 min. After washing in distillate water, sections were drained from the staining solution, quickly dehydrated in 100% ethanol for 20–40 s, dipped in xylene and cover-slipped. Images were acquired by using a Leica DMR microscope (Leica Microsystems, Wetzlar, Germany).

### 4.3. Immunohistochemistry

Dewaxed sections were treated with 1% H_2_O_2_ in PBS for 15 min to inhibit any endogenous peroxidases activity. The slides for Collagen I were treated using a heat-induced antigen retrieval with citrate buffer pH 6.0, at 95 °C for 20 min, followed by washings in PBS. Following the incubation in blocking solution [PBS + 0.2% bovine serum albumin (BSA)] for 1 h, all the slides were incubated with primary antibodies overnight at 4 °C: Rabbit Polyclonal Anti S100 (marker for myelinating and non-myelinating Schwann cells, Agilent-Dako, dilution 1:4000); Goat Anti Collagen I (for the identification of collagen fibers, Southern-Biotech, Birmingham, AL, USA, dilution 1:400); Mouse Anti human Smooth Muscle Actin (for the smooth-muscle cells of the arteries, Clone 1A4 -Agilent Dako, Santa Clara, CA, USA, dilution 1:200). After repeated PBS washing, the sections were incubated with the secondary antibodies for 1 h: goat anti-rabbit IgG (Jackson ImmunoResearch, West Baltimore, PA, USA, dilution 1:300), rabbit anti-goat peroxidase-AffiniPure IgG (Jackson ImmunoResearch, dilution 1:300), kit Advance HRP Rabbit/Mouse (ready to use, Agilent Dako, Santa Clara, CA, USA). Negative controls followed the same protocol, with the omission of the primary antibody. The reaction was then developed with 3,3′-diaminobenzidine (Liquid DAB + substrate Chromogen System kit Dako), stopped with distilled water and counterstained with Toluidine Blue 0.1%.

### 4.4. Transmission Electron Microscopy (TEM)

Two specimens from each of the collected SF samples were randomly selected and fixed in 2.5% glutaraldehyde (Serva Electrophoresis, Heidelberg, Germany) in 0.1 M phosphate buffer pH 7.4, post-fixed in 1% osmium tetroxide (OsO4, Agar Scientific Elektron Technology, Stansted, UK) in 0.1 M phosphate-buffer pH 7.4, dehydrated in a graded ethanol series and then embedded in an Epoxy Embedding Medium Kit (Sigma-Aldrich, St. Gallen, Switzerland). Semithin (0.5 μm) and ultrathin (60 nm) sections were cut with an RMC PowerTome ultramicrotome (Boeckeler Instruments, Tucson, AZ, USA). The semithin sections were stained with 1% Toluidine Blue pH 6 on a hot plate (+80 °C). The ultrathin sections were collected on 300-mesh copper grids and counterstained with 1% uranyl acetate and then with Sato’s lead solution [42]. The specimens were examined using a Hitachi H-300 Transmission Electron Microscope (TEM) (Hitachi, Tokyo, Japan).

### 4.5. Image Analysis: Density of Mast Cells

A Toluidine Blue positive reaction was observed using a Leica DMR microscope (Leica Microsystems, Wetzlar, Germany). To estimate mast cell density in tissue sections, computer-assisted image analysis (see Figure 3) was performed by using the ImageJ software (freely available at http://rsb.info.nih.gov/ij/, accessed on 17 April 2023). Briefly, after shading correction and contrast enhancement, images were converted to 8-bit grey-level images. By conventional thresholding methods [43], stained cell profiles can be easily discriminated. Those belonging to mast cells can then be identified by applying selection criteria based on size and shape, and their density (number per unit area of tissue) estimated. The analysis was performed in 2 slices for each SF sample. The mean and standard deviation of mast cells’ density per mm^2^ of tissue were calculated.

Furthermore, the relative percentage of distribution of mast cells in the connective tissue area, close to the nerves, or in the adipocyte areas, close to the blood vessels, was calculated (% mean ± % standard deviation) in the same samples by manual count using a Leica DMR microscope.

## Figures and Tables

**Figure 1 ijms-24-11599-f001:**
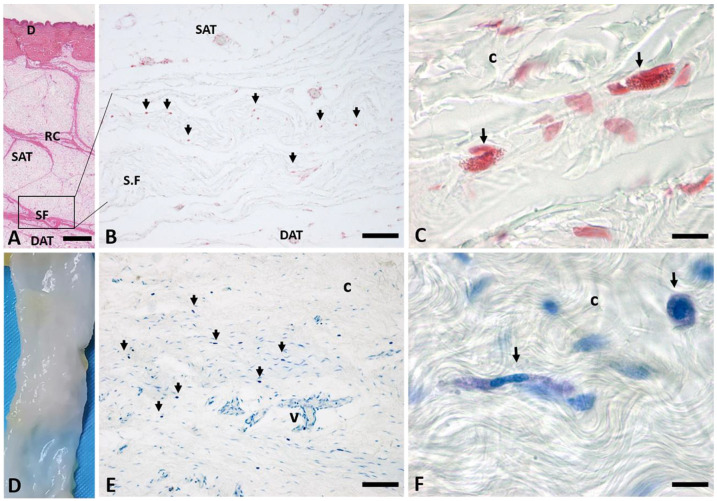
Superficial fascia (SF) layer of the abdomen, stained with Safranin-O (**B**,**C**) and with Toluidine Blue (**E**,**F**). (**A**) Hematoxylin and eosin stain of all the layers of a formalin-fixed sample of the abdominal region, from skin (D: dermis) to the superficial fascia (SF). (**B**,**C**) Safranin-O stainings. Arrows indicate mast cells. The SF highlighted in (**A**) is isolated from the SAT and the DAT (**D**). (**E,F**) Toluidine Blue staining of the tangential section of the flat embedded SF. Arrows indicate mast cells. D: dermis; RC: retinacula cutis superficialis; SAT: superficial adipose tissue; SF: superficial fascia; DAT: deep adipose tissue; c: collagen fibers; v: blood vessel. (**C**,**F**) are obtained using oil-immersion 100× objective. Scale Bars: (**A**) 1.5 mm; (**B**–**E**) 100 μm; (**C**–**F**) 10 μm.

**Figure 2 ijms-24-11599-f002:**
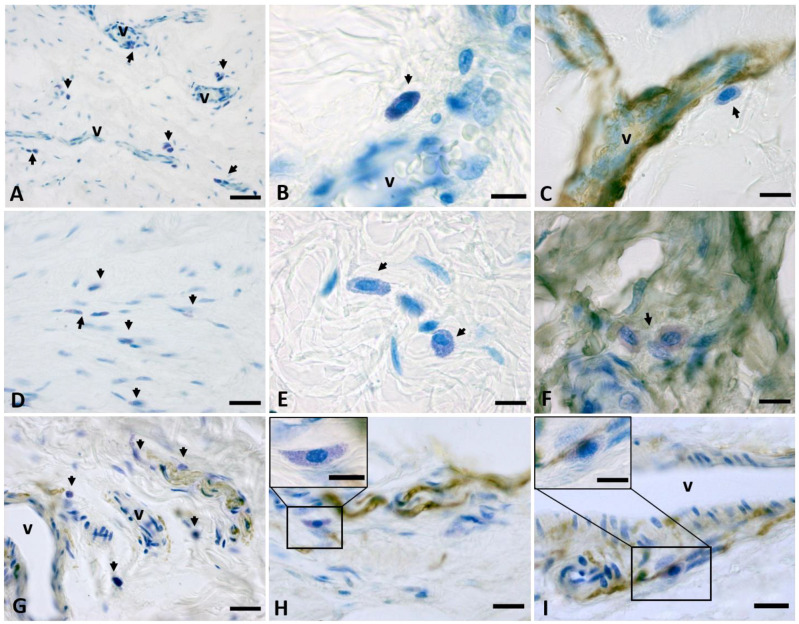
Mast cells localization in Superficial Fascia layer—Toluidine Blue staining. Mast cells are evident close to the blood vessels supplying the SF (**A**–**C**), in the midst of the collagen fibers (**D**–**F**) and close to the S100-positive neuronal elements (**G**–**I**). In panel C are evident the smooth-muscle cells of the arteries, positive to α-sma antibody; in F, the positive immunoreaction for Collagen I. Some erythrocytes are evident in the lumen of some blood vessels (**B**,**C**). Arrows indicate mast cells; v: vessels. (**B**,**C**,**E**,**F**) and insets are obtained using oil-immersion 100× objective. Scale bars: (**A**) 50 μm; (**D**,**G**) 25 µm; (**B**,**C**,**E**,**F**,**H**,**I**) 10 μm; insets: 5 μm.

**Figure 3 ijms-24-11599-f003:**
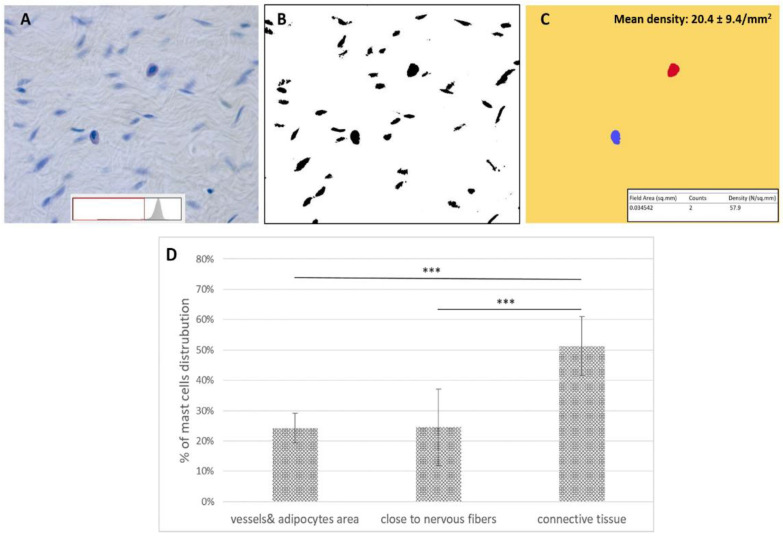
(**A**–**C**) Schematic illustration of the image analysis procedures. (**A**) Microscope image from a tissue section stained with Toluidine Blue (primary magnification ×40). The grey-level histogram is shown on the bottom, together with the threshold applied to discriminate cells. (**B**) Binary image of the stained cell profiles (**C**). From the binary image in (**B**), mast cell profiles can be extracted by selecting those exhibiting a size greater than 800 pixels and a circular shape factor greater than 0.80 [13]. The estimated density is shown in the inset, and the mean density of all the analyzed images is indicated in the upper right. (**D**) Relative percentage of mast cells distribution in the connective tissue area, close to the nervous fibers, or in the adipocytes areas/close to the blood vessels (% mean ± % standard deviation) in SF samples. One-way ANOVA test showed very statistically significant differences (*** *p*-value < 0.001) between the distribution of mast cells in the connective tissue area with respect to the two other areas (close to the nervous fibers and between vessels and adipocytes), which instead show a similar density (with no statistically significant difference, *p*-value > 0.05).

**Figure 4 ijms-24-11599-f004:**
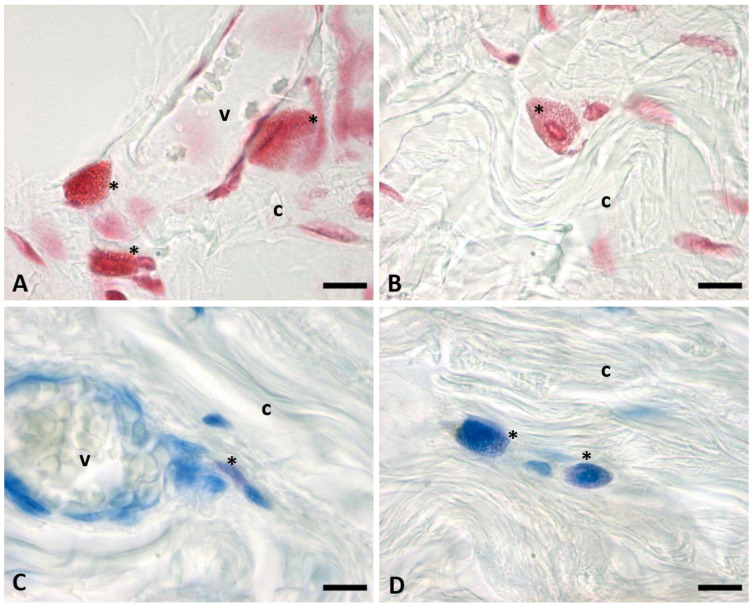
Mast cells close to the blood vessels (**A**,**C**) and in the connective tissue of SF (**B**,**D**). (**A**,**B**) are stained by Safranin-O; (**C**,**D**) are stained by Toluidine Blue. Images are obtained using oil-immersion 100× objective. *: mast cells; c: collagen; v: vessels. In the lumen of the blood vessels (v) are evident some erythrocytes. Scale bars: 10 μm.

**Figure 5 ijms-24-11599-f005:**
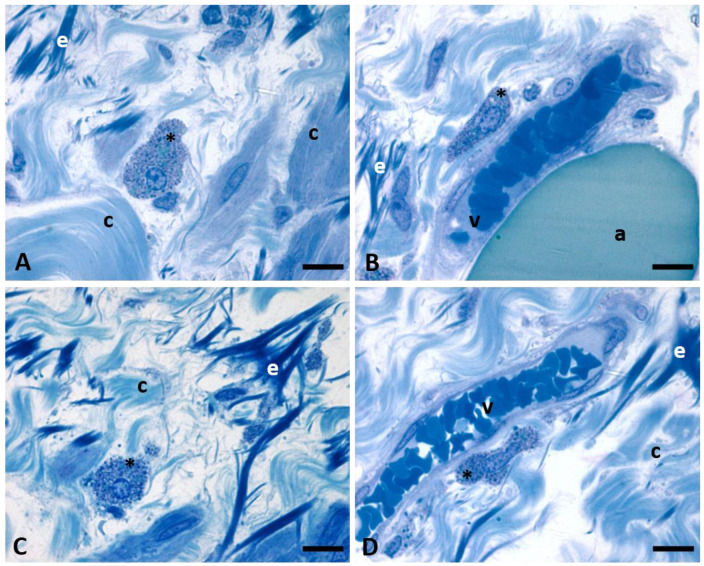
Semithin sections of SF stained with 1% Toluidine Blue. Mast cells (*) are evident in the connective tissue (**A**,**C**) and close to the blood vessels (**B**,**D**). In the cytoplasm of the mast cells (*) are evident the granules. Between the collagen fibers (c) are visible some elastic fibers (e), blood vessels (v) with erythrocytes in the lumen, and adipocytes (a). Images are obtained using oil-immersion 100× objective. Scale bars: 10 μm.

**Figure 6 ijms-24-11599-f006:**
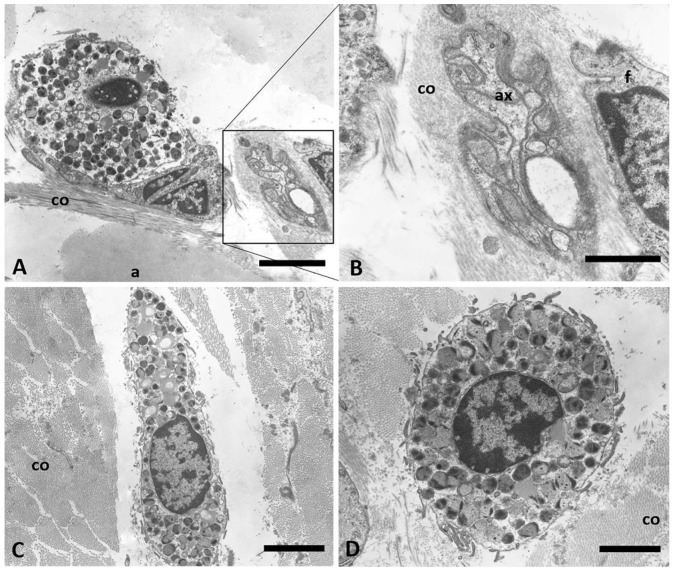
TEM images of mast cells. (**A**,**C**,**D**) show mast cell in detail, with a big nucleus and evident cytoplasmic granules. Mast cells are in the midst of collagen fibers (co) longitudinally (**A**) or transversally (**C**,**D**) arranged, and close to the nerves (inset of panel (**A**), and panel (**B**)—ax: unmyelinated axon of the nerve). a: adipocyte; f: fibroblast. Scale bars: (**A**,**C**) 4 µm; (**B**–**D**) 2 μm.

## Data Availability

The original contributions presented in this study are included in the article; further inquiries can be directed to the corresponding author.

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
