# Peer review of "Detection of Mast Cells in Human Superficial Fascia"

_ijms, 2023, doi:10.3390/ijms241411599_

Round 1
Reviewer 1 Report
The manuscript reports the occurrence of mast cells in superficial fascia from abdominal skin, which is worth-studying. It has some merit for publication, pending to the corrections as detailed below:
Results: ---blue/purple stained cytoplasmic granules. Please add metachromatic granules in description.
Fig. 1. -----stained with Safranin-O (B, C) or Toluidine Blue (E, F). Replace or with and. Mention what is S.F? Fig. A is highly contrasted, making it difficult to see the all essential histological features. What is the magnification of this image? Insert the scale bar on it.
Figure 2. ----close to the nervous structure S100-positive (C, F). Better, close to the S100-positive neuronal elements. While I agree that scale bar values for C-F could be 10 μm, the same for the insets should be different [<10 μm]. Also, if any image is taken using 100X oil-immersion objective, as is the case, this must be mentioned in legends.
secretory lysosomes, should be secretory granules
Fig. 6D. Mention that surface projections are evident.
Discussion: high-density [20.5 ± 4/mm2] presence of mast cells in SF…. Please add few comparisons with other works (with similar or different tissues) to validate it. Correct stimulatin
A line must be mentioned whether the described mast cells could be chymase or tryptase positive.
Methods: 5’, 10’ ----should be 5 min, 10 min.
Immunohistochemistry--- secondary Goat anti rabbit, should be secondary goat anti-rabbit IgG. Mention the dilution used for it.
Transmission Electron Microscopy (TEM)--- Mention the buffer pH. Sato's 281 lead solution, mention the compound present in it.
Please avoid using capitalisation of words (e.g., Acetic Acid, should be acetic acid).
The authors are advised to consult a paper in Indian J Plast Surg. 2010; 43: 135–140, where mast cells have been described in deep fascia.
Author Response
The manuscript reports the occurrence of mast cells in superficial fascia from abdominal skin, which is worth-studying. It has some merit for publication, pending to the corrections as detailed below:
Results: ---blue/purple stained cytoplasmic granules. Please add metachromatic granules in description. We have added as suggested.
Fig. 1. -----stained with Safranin-O (B, C) or Toluidine Blue (E, F). Replace or with and. Mention what is S.F? Thank you, we modified.
Fig. A is highly contrasted, making it difficult to see the all essential histological features. What is the magnification of this image? Insert the scale bar on it. We modified the Panel A of Figure 1, modifying the contrast and adding the scale bar.
Figure 2. ----close to the nervous structure S100-positive (C, F). Better, close to the S100-positive neuronal elements. While I agree that scale bar values for C-F could be 10 μm, the same for the insets should be different [<10 μm]. Also, if any image is taken using 100X oil-immersion objective, as is the case, this must be mentioned in legends.
We thank the reviewer for the suggestion: we followed the indications, added in the legends when we used a 100X oil-immersion objective, and modified the scale bars of the insets
(thank you, it was our mistake).
secretory lysosomes, should be secretory granules. Thank you, changed.
Fig. 6D. Mention that surface projections are evident. We have mentioned as suggested by the reviewer.
Discussion: high-density presence of mast cells in SF…. Please add few comparisons with other works (with similar or different tissues) to validate it.
Thank you for the precious suggestion: we have added a new paragraph with some references for the comparison with the distribution of mast cells in other human tissues: “[..] the distribution of mast cells changes according the age (it is higher in children with respect to adults) and show some heterogeneity, considering also the fact that mast cells are frequently sparse and can be easily missed during the manual count [24]. However, we found a density equal to 20.4 ± 9.4/mm2, which is in line with what has already been shown: the normal range of mast cells in adult skin (mean age 43.8±16.1 years) was established for some Authors below 40/mm2 [24]. Other works showed how the density increases in the skin of patients with early stages of scleroderma (111 +/- 28 cells/mm2) with respect to normal controls (50±14 cells/mm2) [25]. It has to be considered that the differences among regions might be due to region-specific mast cell roles: some Authors, for instance, showed a mast cell richness in the muscle coat in human ileocecal region, especially in the inner circular muscle layer (density equal to 72.83/mm2), that might be important in regulating its motility [26].”
Correct stimulatin. We have corrected.
A line must be mentioned whether the described mast cells could be chymase or tryptase positive. We agree with the reviewer and we added this sentence in the Discussion: “Furthermore, future studies will shed light to the content of the granules of mast cells: a further analysis will be fundamental for the classification of these mast cells into the two classical subsets, tryptase-positive and chymase-positive [39].”
Methods: 5’, 10’ ----should be 5 min, 10 min. We have modified.
Immunohistochemistry--- secondary Goat anti rabbit, should be secondary goat anti-rabbit IgG. Mention the dilution used for it. We have modified.
Transmission Electron Microscopy (TEM)--- Mention the buffer pH. We added that the pH is equal to 7.4.
Sato's 281 lead solution, mention the compound present in it. The solution contains Lead Nitrate/Lead Citrate/Lead Acetate/Sodium Citrate: we added a specific reference about it (ref. n. 42)
Please avoid using capitalisation of words (e.g., Acetic Acid, should be acetic acid). Thank you, we modified.
The authors are advised to consult a paper in Indian J Plast Surg. 2010; 43: 135–140, where mast cells have been described in deep fascia.
Authors thank the reviewer for this suggestion. We have added this reference and other new references indicating the description of mast cells in human deep fascia, and in superficial fascia of rats. The new paragraph is: “A first evidence of mast cells in human fascia was showed in the deep fascia of the lower limbs [13] as well as in human fascia lata [14], by Transmission Electron Microscopy. Furthermore, a recent work highlighted the presence of mast cells in the superficial fascia of rats, closely associated with fascial adipogenic progenitors and mature adipocytes: the numerous heparin-loaded granules released from mast cells were distributed around fascial preadipocytes, indicating that mast cells could serve as endogenous physiological factors to initiate fascial adipogenesis [15]. However, this work constitutes the first histological investigation of the presence of active mast cells in the human superficial fascia, ubiquitously distributed, especially between the collagen fibers of the tissue.”
Reviewer 2 Report
Fede C. and colleagues, in their manuscript entitled “MAST CELLS IN SUPERFICIAL FASCIA AS MODULATORS OF TISSUE HEALING”, show for the first time the presence of Mast cells in the superficial fascia of the abdominal skin in humans. The authors used complementary specific stains and TEM to show the presence and distribution of mast cells within the superficial fascia. Due to the pro-inflammatory function of mast cells and the surging role of the superficial fascia in repair processes, this work would represent a primordial study on the immunological niche of the human superficial fascia. Nonetheless, changes in certain interpretations, additional analysis, and a more careful discussion are necessary for this study before been accepted for publication.
Major comments:
- Although safranin and toluidine stains can complementarily indicate the presence of Mast cells, what these stains do is to dye basic cytoplasmic granules, which are not entirely restricted in Mast cells (basophils also have basic granules). Even though TEM can be used to discriminate between other cell types containing granules, most of the images and, more importantly, the quantifications presented were performed with safranin and toluidine stains, in which the distinction of true mast cells from the background rely entirely on the observer.
A more objective detection/quantification method should be considered. IHC for mast-specific markers (e.g. chymase) would provide a definitive proof of the nature of the cells observed in this study. Image segmentation methods could also be performed in toluidine/ safranin-stained images to extract the more intensively dyed cells for a more objective quantification. - Figures 2, 4 and 5 emphasize mast cell localization near blood vessels and ECM without showing objective detection of the aforementioned tissue structures. Similar to point 1, determination of blood vessels and ECM fibers rely solely on the authors’ empiric experience. Similar to the detection of nerves with the S100 antibody, the authors should perform IHC for markers of blood vessels (e.g. PECAM1) and collagen fibers to sustain their conclusions.
- Figure 3 summarizes the main observations of this work, yet the numbers are presented in a vacuum that impedes understand the relevance of the data. Are these numbers a likely reflection of the potential role of Mast cells in the superficial fascia? How do they compare to other tissues where the role of Mast cells has been proven? This is particularly relevant as many of the authors’ conclusions in the discussion section branch from these quantifications (see minor points 2-4). The authors should provide additional quantifications from other tissues, preferably dermis, to better contextualize the relevance of the presented data.
- Please also provide statistical tests of the data in figure 3 to sustain the conclusions of a preferred localization of mast cells within the connective tissue.
- Main title of the work is misleading as no evidence of a modulatory role in situ was provided. A more fitting title (e.g. “DETECTION OF MAST CELLS INHUMAN SUPERFICIAL FASCIA”) should be considered.
- The authors need to include and discuss more relevant studies so the reader can have a more fitting perspective of the current knowledge of mast cells in fascial tissues. For instance, Dawidowicz and colleagues showed the presence of Mast cells in human fascia lata in 2015 (PMID: 26311620) and works by Xu Guoheng’s group showed the presence of mast cells, their strong association with blood vessels, and their adipogenic role in rat superficial fascia (PMIDs: 36452040, 34389520, 31549232).
Minor comments:
- Panel B of figure 6 is flipped and does not correspond to the low mag image on the side.
- Statements in line 85 “Mast cells are uniformly distributed in the SF tissue” and later in line 103 “Most of these cells (51% ± 9.7%) are in the connective tissue area…” are contradictory. If mast cells are mostly amidst connective tissue and not near vessels or nerves, then the distribution is not uniform. Please rephrase.
- Line 169: “The finding of high-density presence of mast cells in SF, distributed all over the tissue but above all in the dense connective tissue itself, can help to confirm that the SF participates in regulating and healing processes.” The authors cannot conclude that the density is high if not compared to something else (Major point 3). Furthermore, evidence of presence cannot “confirm” their role in a healing or any process. Please rephrase.
- Line 194: “These evidences can be supported by our findings of a high density of mast cells in direct contact with the fascial fibroblast, opening the possibility of the involvement of the SF in regulating and modulating functions”. Direct contact of mast cells to fibroblasts was only shown in the TEM images in figure 6 and from those it is not possible to infer “high density” as this could be a rare observation. Please rephrase, otherwise quantifications of the distance between fibroblasts and mast cells from several images should be provided.
the quality of English language could be improved.
Author Response
Comments and Suggestions for Authors
Fede C. and colleagues, in their manuscript entitled “MAST CELLS IN SUPERFICIAL FASCIA AS MODULATORS OF TISSUE HEALING”, show for the first time the presence of Mast cells in the superficial fascia of the abdominal skin in humans. The authors used complementary specific stains and TEM to show the presence and distribution of mast cells within the superficial fascia. Due to the pro-inflammatory function of mast cells and the surging role of the superficial fascia in repair processes, this work would represent a primordial study on the immunological niche of the human superficial fascia. Nonetheless, changes in certain interpretations, additional analysis, and a more careful discussion are necessary for this study before been accepted for publication.
Major comments:
- Although safranin and toluidine stains can complementarily indicate the presence of Mast cells, what these stains do is to dye basic cytoplasmic granules, which are not entirely restricted in Mast cells (basophils also have basic granules). Even though TEM can be used to discriminate between other cell types containing granules, most of the images and, more importantly, the quantifications presented were performed with safranin and toluidine stains, in which the distinction of true mast cells from the background rely entirely on the observer.
A more objective detection/quantification method should be considered. IHC for mast-specific markers (e.g. chymase) would provide a definitive proof of the nature of the cells observed in this study. Image segmentation methods could also be performed in toluidine/ safranin-stained images to extract the more intensively dyed cells for a more objective quantification.
We thank the reviewer for the optimal indication and suggestion: surely, in the next future it will be fundamental to add a specific staining for mast cells to divide them in the two typical subsets, tryptase/chymase-positive. However, we have used two histological stains and the Transmission Electron Microscopy technique in this preliminary study: especially the TEM technique is an objective method to provide a definite proof of the nature of these cells. Another work in the human deep fascia of the lower limb, published in 2010, demonstrated the presence of mast cells only by TEM: Authors added in the discussion this specific reference (Indian J Plast Surg. 2010; 43: 135–140).
We agree that a more objective quantification could be useful to discriminate the mast cells: for that reason, we followed the reviewer’s suggestion and substitute the manual count of the density with a computer-assisted image analysis. Anyway, the mean density showed only a small variation with respect to the previous manual count, confirming the solidity of our distribution analysis. The new methods are:
“Toluidine Blue positive reaction was observed using a Leica DMR microscope (Leica Microsystems,Wetzlar, Germany). To estimate mast cell density in tissue sections, computer-assisted image analysis (see Fig.3) was performed by using the ImageJ software (freely available at http://rsb.info.nih. gov/ij/). Briefly, after shading correction and contrast enhancement, images were converted to 8-bit grey-level images. By conventional thresholding methods [J.C. Russ, The Image Processing Handbook, 6, Taylor & Francis, Boca Raton, 2011] stained cell profiles can be easily discriminated. Those belonging to mast cells can then be identified by applying selection criteria based on size and shape and their density (number per unit area of tissue) estimated. The analysis was performed in 2 slices for each SF sample. Mean and standard deviation of mast cells density per mm2 of tissue were calculated.
Furthermore, the relative percentage of distribution of mast cells in the connective tissue area, close to the nerves, or in the adipocyte areas close to the blood vessels was calculated (% mean ± % standard deviation) in the same samples by manual count using a Leica DMR microscope.”
The new Figure 3 is now uploaded with this new legend:
Figure 3. A-C: Schematic illustration of the image analysis procedures. A: Microscope image from a tissue section stained with toluidine blue (primary magnification x40). The grey level histogram is shown on the bottom together with the threshold applied to discriminate cells. B: Binary image of the stained cell profiles C. From the binary image in B mast cell profiles can be extracted by selecting those exhibiting a size higher than 800 pixels and a circular shape factor greater than 0.80 [Neal FB, Russ JC. Measuring shape. CRC Press, Boca Raton, 2012]. The estimated density is shown in the inset, and the mean density of all the analysed images is indicated in the upper right.
D: relative percentage of mast cells distribution in the connective tissue area, close to the nervous fibers, or in the adipocytes areas/close to the blood vessels (% mean ± % standard deviation) in SF samples. One-way Anova test showed very statistically significant differences (p-value<0.001) between the distribution of mast cells in the connective tissue area with respect to the two other areas (close to the nervous fibers and between vessels and adipocytes), which instead show a similar density (with no statistically significant difference, p-value>0.05).
2. Figures 2, 4 and 5 emphasize mast cell localization near blood vessels and ECM without showing objective detection of the aforementioned tissue structures. Similar to point 1, determination of blood vessels and ECM fibers rely solely on the authors’ empiric experience. Similar to the detection of nerves with the S100 antibody, the authors should perform IHC for markers of blood vessels (e.g. PECAM1) and collagen fibers to sustain their conclusions.
Authors followed this good reviewer’s comment, adding the α-sma marker for the smooth muscle cells of the arteries and collagen-I marker for the collagen fibers of the superficial fascia. By this way, we provide a definitive proof of the tissue structures and of the distribution of mast cells close to these structures. Authors provided a new version of Figure 2 with these new details. In addition, the TEM technique provides clear evidence of the structures of the tissue (adipocytes, collagen fibers, vessels, nerves).
3. Figure 3 summarizes the main observations of this work, yet the numbers are presented in a vacuum that impedes understand the relevance of the data. Are these numbers a likely reflection of the potential role of Mast cells in the superficial fascia? How do they compare to other tissues where the role of Mast cells has been proven? This is particularly relevant as many of the authors’ conclusions in the discussion section branch from these quantifications (see minor points 2-4). The authors should provide additional quantifications from other tissues, preferably dermis, to better contextualize the relevance of the presented data.
Authors added a specific paragraph in the discussion to compare the density of mast cells found in this work with other analysis of mast cells distribution, in skin and other tissues, related also with age and some pathologies. The new paragraph is the following: “However, the distribution of mast cells changes according the age (it is higher in children with respect to adults) and show some heterogeneity, considering also the fact that mast cells are frequently sparse and can be easily missed during the manual count [22]. However, we found a density equal to 20.4 ± 9.4/mm2, which is in line with what has already been shown: the normal range of mast cells in adult skin (mean age 43.8±16.1 years) was established for some Authors below 40/mm2 [22]. Other works showed how the density increases in the skin of patients with early stages of scleroderma (111 +/- 28 cells/mm2) with respect to normal controls (50±14 cells/mm2) [23]. It has to be considered that the differences among regions might be due to region-specific mast cell roles: some Authors, for instance, showed a mast cell richness in the muscle coat in human ileocecal region, especially in the inner circular muscle layer (density equal to 72.83/mm2), that might be important in regulating its motility [24].”
As described also by cited references, the density of mast cells shows high heterogeneity, according the age, the pathologies, the anatomical region, but also among regions of the same anatomical area, according to the specific function and the high potentiality of the roles of mast cells.
4. Please also provide statistical tests of the data in figure 3 to sustain the conclusions of a preferred localization of mast cells within the connective tissue.
We thank the reviewer for the indication: we have added the one-way ANOVA test to compare the density of mast cells in the connective tissue area, in the vessels and adipocyte area, and close to the nervous fibers. We found that the major localization of mast cells is between the collagen fibers, with a statistically significant difference (p value <0.001) with respect the two other areas, which show a similar density (with no statistically significant difference, p-value>0.05). The statistics was added in the Figure 3 and in the text.
5. Main title of the work is misleading as no evidence of a modulatory role in situ was provided. A more fitting title (e.g. “DETECTION OF MAST CELLS INHUMAN SUPERFICIAL FASCIA”) should be considered.
We modified the title according the indication of the reviewer.
6. The authors need to include and discuss more relevant studies so the reader can have a more fitting perspective of the current knowledge of mast cells in fascial tissues. For instance, Dawidowicz and colleagues showed the presence of Mast cells in human fascia lata in 2015 (PMID: 26311620) and works by Xu Guoheng’s group showed the presence of mast cells, their strong association with blood vessels, and their adipogenic role in rat superficial fascia (PMIDs: 36452040, 34389520, 31549232).
We thank the reviewer for this comment. We added new references in the Discussion, the new paragraph is: “A first evidence of mast cells in human fascia was showed in the deep fascia of the lower limbs [13] as well as in human fascia lata [14], by Transmission Electron Microscopy. Furthermore, a recent work highlighted the presence of mast cells in the superficial fascia of rats, closely associated with fascial adipogenic progenitors and mature adipocytes: the numerous heparin-loaded granules released from mast cells were distributed around fascial preadipocytes, indicating that mast cells could serve as endogenous physiological factors to initiate fascial adipogenesis [15]. However, this work constitutes the first histological investigation of the presence of active mast cells in the human superficial fascia, ubiquitously distributed, especially between the collagen fibers of the tissue.”
Minor comments:
- Panel B of figure 6 is flipped and does not correspond to the low mag image on the side.
We agree with the reviewer: panel B was modified.
2. Statements in line 85 “Mast cells are uniformly distributed in the SF tissue” and later in line 103 “Most of these cells (51% ± 9.7%) are in the connective tissue area…” are contradictory. If mast cells are mostly amidst connective tissue and not near vessels or nerves, then the distribution is not uniform. Please rephrase.
Thanks for the observation: we changed “uniformly” with “ubiquitously”.
3. Line 169: “The finding of high-density presence of mast cells in SF, distributed all over the tissue but above all in the dense connective tissue itself, can help to confirm that the SF participates in regulating and healing processes.” The authors cannot conclude that the density is high if not compared to something else (Major point 3). Furthermore, evidence of presence cannot “confirm” their role in a healing or any process. Please rephrase.
Authors rephrased with this new version: “The finding of the presence of active mast cells in SF, distributed all over the tissue but above all in the dense connective tissue itself, with a very statistically significant difference (p<0.001) with respect to both nervous fibers and vessels/adipocytes areas, can help to suppose that the SF participates in regulating and healing processes.”
Moreover, according to the major point 3, we added a comparison with other works in which Authors analyzed the density of mast cells in human tissues.
4. Line 194: “These evidences can be supported by our findings of a high density of mast cells in direct contact with the fascial fibroblast, opening the possibility of the involvement of the SF in regulating and modulating functions”. Direct contact of mast cells to fibroblasts was only shown in the TEM images in figure 6 and from those it is not possible to infer “high density” as this could be a rare observation. Please rephrase, otherwise quantifications of the distance between fibroblasts and mast cells from several images should be provided.
The distribution of mast cells close to the fascial fibroblast is evident in the TEM image, but it is also shown in Figures 1 and 4 (we added this point in the results). We removed from the sentence the adjective “high” density.
